# Implant-Supported Cantilever Fixed Partial Dentures in the Posterior Region: A Systematic Review and Meta-Analysis on Survival Outcomes

**DOI:** 10.3390/ma18204704

**Published:** 2025-10-14

**Authors:** Fernanda L. Vieira, Maria N. B. Fontenele, Leticia M. de Souza, Thayná S. Berteli, Renata C. S. Rodrigues, Ester A. F. Bordini, Joel F. Santiago Junior

**Affiliations:** Department of Dental Materials and Prosthodontics, School of Dentistry of Ribeirão Preto, University of São Paulo, Ribeirão Preto 14040-904, SP, Brazil; fernandaleal.v@usp.br (F.L.V.); natallybelchior@usp.br (M.N.B.F.); leticiamoreira@usp.br (L.M.d.S.); thayna.berteli@usp.br (T.S.B.); renata@forp.usp.br (R.C.S.R.); esterbordini@usp.br (E.A.F.B.)

**Keywords:** systematic review, dental implants, cantilever

## Abstract

Implant-supported partial dentures with cantilever extensions (ISPDCs) present significant biomechanical challenges when rehabilitating partially edentulous patients, especially in the posterior region, where higher complication rates are often reported. This systematic review aimed to evaluate the complications, survival rates, and marginal bone loss associated with ISPDCs in posterior areas. The review protocol was registered with the PROSPERO database (CRD42024606201) and was conducted in accordance with PRISMA-P guidelines. A comprehensive search was performed across 10 databases for studies published up until 28 January 2025. Out of an initial 2142 records, 11 clinical studies met the inclusion criteria. The analysis showed a low failure rate for both prostheses and implants, at 1% (95% confidence interval: 0–3%) across the studies evaluated. However, noteworthy complications arose from biological factors as well as technical issues. Complications related to the prosthesis and abutment occurred in 14% of cases (95% CI: 5–26%), while loss of retention was noted in 13% of cases (95% CI: 7–21%). These technical issues were particularly more frequent in extensions greater than 7 mm. A significant difference was observed in marginal bone loss when comparing the final and initial measurements to the cantilever (*p* < 0.0001), which was influenced by the study design. Factors such as the type of occlusal veneering material, study design related to biological complications, the number of implants, marginal bone loss (between adjacent and distant sites), and the retention system did not significantly affect the complication rate (*p* > 0.05). The certainty of evidence for all primary outcomes was rated as low due to study limitations, heterogeneity, and risk of bias. In conclusion, ISPDCs show favorable survival outcomes when supported by adequate planning and clinical monitoring, although longer extensions require more cautious case selection and prosthetic design.

## 1. Introduction

The success of implant-supported rehabilitations is closely linked to meticulous execution and accurate case planning. However, in posterior regions, limited bone volume and reduced prosthetic space often restrict implant placement, making inserting one implant per missing tooth unfeasible. In such scenarios, prosthetic rehabilitation using pontics and cantilever extensions represents a viable alternative [1,2,3].

However, the use of implant-supported partial dentures with cantilever extensions (ISPDCs) presents a clinical challenge, as it involves increased complexity in treatment planning, implant placement, selection, and prosthesis fabrication [4,5]. Furthermore, this rehabilitation approach presents unfavorable biomechanical characteristics, such as lever forces inherent to its design and increased incidence of non-axial loading, especially when compared to implant-supported partial dentures without cantilever extensions [6,7,8].

The literature indicates that most complications related to implant-supported prostheses are biomechanical [9,10], highlighting the need for further investigations into ISPDCs that specifically address these factors. A recent systematic review evaluating implant-supported prostheses placed in both anterior and posterior regions reported a lower survival rate for ISPDCs in the posterior region. Conversely, greater marginal bone loss was observed around implants in the cantilever group within the anterior region. These differences may be directly associated with the variation in force intensity across regions, as well as the predominance of compressive forces in the posterior and shear stresses in the anterior region, respectively [11]. Nonetheless, the literature still lacks comprehensive analyses addressing technical and biological complications, cantilever length, and marginal bone loss concerning implant positioning.

A systematic review specifically addressing implants placed in the posterior region is needed due to the higher masticatory forces in this area and the associated risk of treatment failure. Therefore, this study aims to evaluate the survival rate, marginal bone loss, and the incidence of both technical and biological complications related to implants in the posterior region.

## 2. Materials and Methods

This study was conducted following the guidelines established by the Cochrane Collaboration and the PRISMA criteria [12,13,14]. It is registered in the PROSPERO database (CRD42024606201). The systematic review was structured based on the PICO framework, which includes the resulting components: Population: individuals requiring oral rehabilitation with dental implants; Intervention: rehabilitation using implant-supported partial dentures with cantilever extensions in the posterior region; Comparison: patients rehabilitated with implant-supported fixed partial dentures without cantilever extensions in the posterior region; Outcome: assessment of implant and prosthesis survival rates, technical and biological complications, and marginal bone loss associated with both treatment approaches.

The selection of studies followed these inclusion criteria: articles published in English, Portuguese, or Spanish; prospective, retrospective, randomized, or non-randomized clinical trials with a sample size greater than five participants per group; studies evaluating the survival rate and complications of ISPDCs in the posterior region. The exclusion criteria included duplicate publications; articles with incomplete data; studies that did not clearly distinguish between groups with and without cantilever; studies that did not specify the implant placement region (anterior/posterior); studies evaluating full-arch prostheses; and studies assessing tooth-supported prostheses.

The following databases were searched: Medline/PubMed, Cochrane Library, EMBASE, SciELO, EBSCO, LILACS, BVS, Web of Science, Scopus, and Google Scholar. The search included articles published up to 28 January 2025, using the keywords “dental implants”, “fixed partial dentures”, and “cantilever”, combined with the Boolean operator “AND”. The retrieved articles were imported into Rayyan^®^ (Rayyan Systems Inc., Cambridge, MA, USA), a web- and mobile-based application developed for systematic reviews. Duplicate records were removed within the platform, and article selection was carried out by reviewers who were blinded to the study’s objectives. The search process, study selection, and data extraction were conducted by previously calibrated researchers (F.L.V., M.N.B.F., and L.M.S). In cases of disagreement, a fourth reviewer (J.F.S. Jr.) was consulted to resolve the conflict.

Data were collected using an Excel spreadsheet (Excel^®^, Microsoft, Redmond, WA, USA) are summarized in (Section 3). The collected variables included the following: author, year of publication, study design, follow-up duration, mean patient age, total number of patients, number of implants, implant location (maxilla/mandible), implant brand, number of prostheses, number of prostheses with cantilever extensions, prosthetic material, cantilever length, cantilever position (mesial/distal), type of prosthesis retention (cemented/screw-retained), as well as implant and prosthesis survival rates and reported complications. In cases where data reliability was uncertain, clarification was obtained by contacting the corresponding author of the respective study [15].

The clinical studies included in this analysis were classified according to the study design hierarchy established by the National Health and Medical Research Council (NHMRC) [16]. The risk of bias in randomized controlled trials (RCTs) was evaluated using the Cochrane Collaboration’s risk of bias tool. This tool assesses various domains, including selection bias (random sequence generation), deviations from intended interventions, missing outcome data, outcome measurement bias, and selective reporting bias [17]. For non-RCT studies, the Newcastle-Ottawa scale was used, which is based on three major components: selection, comparability, and outcome or exposure [18]. According to the Newcastle–Ottawa Scale, a maximum of nine stars can be awarded to a study, with higher scores indicating greater methodological quality. Studies that received five stars or fewer were considered to be at high risk of bias, whereas those awarded six stars or more were classified as having a low risk of bias [19,20].

The quantitative data extracted from the included studies were organized into tables to facilitate the analysis of the overall mean with a 95% confidence interval (CI). The relative weight of each study was calculated for the meta-analysis of dichotomous outcomes. Survival and complication rates were analyzed based on the number of events and the total number of implants or implant-supported prostheses. Continuous outcomes were analyzed using the mean difference (MD) with corresponding 95% CIs. Freeman-Tukey double arcsine transformation was used for summary measure. A *p*-value of less than 0.05 was considered statistically significant for all analyses. Data analysis and graph generation were performed using an online meta-analysis tool [21].

The primary objective of this study was to quantify the survival and complication rates of implants and ISPDCs in the posterior region. Secondary outcomes included evaluating complications related to the prosthesis or prosthetic abutments, loss of retention, cantilever length, biological complications, retention systems, types of studies included, occlusal veneering materials, and marginal bone loss. Implant survival was defined by the absence of radiolucency around the implant, clinical stability, and the absence of suppuration or pain. Prosthesis survival was characterized by the absence of fractures in the occlusal material, prosthetic infrastructure, or components. Prosthetic failure was defined as the requirement for restoration replacement. Complications were categorized as either technical (ceramic chipping, screw locking, and abutment fracture, loss of retention, including screw loosening, decementation, and loss of retention or biological (such as peri-implant mucositis, peri-implantitis, and gingival discomfort) [2,3,15,22].

A random-effects model was applied for all statistical analyses, and the inverse variance method was applied. Heterogeneity was considered significant at *p* < 0.1 and was assessed using the Q (χ^2^) test, DerSimonian-Laird method, and I^2^ statistic. The I^2^ value was utilized to measure heterogeneity, with values above 75% (on a scale: 0–100%) indicating substantial heterogeneity [9,23]. The certainty of evidence for each outcome was evaluated using the Grading of Recommendations Assessment, Development, and Evaluation (GRADE) approach. The evaluation considered several domains including risk of bias, inconsistency, indirectness, imprecision, and publication bias. The overall certainty of the evidence was categorized as high, moderate, low, or very low to allow for a more nuanced interpretation of the findings. A summary of the evidence was created using the GRADEpro Guideline Development Tool (www.gradepro.org, accessed on 30 March 2025).

## 3. Results

The search strategy yielded a total of 2142 records, from which 1034 duplicates were eliminated. After screening the titles and abstracts, 26 studies were initially selected for full-text review based on their relevance to the clinical evaluation of implant-supported partial dentures with cantilever extensions (ISPDCs). However, nine of these studies were later excluded after a thorough analysis of the full-texts. These excluded studies assessed prostheses in both anterior and posterior regions but failed to clearly distinguish the complications specific to each location. Additionally, two studies did not specify the region of implant placement (anterior/posterior); one focused exclusively on anterior prostheses; two were published in languages outside the eligibility criteria; and one article could not be retrieved in full.

The reasons for exclusion, along with the relevant references, are presented in the Appendix A. As a result, 11 studies met all inclusion and exclusion criteria and were included in the final analysis [2,3,15,22,24,25,26,27,28,29,30]. All articles analyzed in this review were published in English. This process is described in Figure 1.

The studies included in this analysis were published between 2008 and 2023, and key information is summarized in Table 1. Out of the 11 studies, six were retrospective [2,3,24,25,26,27], three were prospective [15,28,29], and two were randomized clinical trials [22,30]. The average follow-up period across these studies was 56.36 months, with durations ranging from a minimum of 24 months [25,28] to a maximum of 228 months [27].

A total of 360 patients were included in the studies, with a mean age of 59.5 years (range: 18–89 years) [27,29]. In total, 291 cantilevered partial dentures and 78 non-cantilevered prostheses were evaluated. Of these, 198 prostheses were placed in the maxilla and 135 in the mandible; however, one study [22] did not specify the arch in which the implants were placed. The number of units per prosthesis varied from one to four, supported by a minimum of one and a maximum of two implants.

A total of 482 implants were evaluated, with lengths ranging from 6 to 15 mm [22,29] and diameters between 3.3 and 7 mm [2,26]. The prosthetic materials utilized included metal–ceramic [2,3,22,24,27,29], metal–composite [28], metal crowns [30], glass fiber–reinforced resin [28], ceramic-veneered gold alloy [2], monolithic zirconia [15], zirconia-coated metal frameworks [3,15], and ceramic-veneered zirconia frameworks [3]. However, some studies did not provide enough detail to clearly identify the materials used in the fabrication of the prostheses.

The retention methods for prostheses varied across the studies. Some utilized cemented prostheses [2,3,24,27,28,30], while others opted for screw-retained prostheses [3,15,22,24,28,29]. Additionally, two studies did not specify the type of retention used [25,26]. The reported lengths of the cantilevers ranged from 3 mm to 12 mm [28,30]; however, five studies [2,3,22,24,25] did not provide information on cantilever length. Of the cantilevered prostheses, 148 were designed with mesial extensions and 143 with distal extensions. The reported technical complications included screw loosening [3,15,22,26,27,28,29,30], decementation [2,3], chipping or fracture of the ceramic veneering [2,3,15,22,27,29], screw blockage [26], screw fracture [26], abutment fracture [26], implant loss [2,22,26,29], and prosthesis fracture [2,26].

The biological complications reported included discomfort related to gingival inflammation [30], peri-implant mucositis [3,15,22,27], and peri-implantitis [2,3,26,28]. Marginal bone loss was inconsistently reported across studies, with considerable variation in data presentation methods, which limited comparative analysis. Overall, marginal bone loss around implants supporting cantilever extensions ranged from 0.1 to 1.55 mm [24,30]. Nevertheless, all studies included in the analysis concluded that the observed bone loss did not contraindicate the use of ISPDCs.

### 3.1. Meta-Analysis

#### 3.1.1. Implant Failure in Cantilevered Prostheses

Ten studies assessed a total of 293 dental implants placed to support cantilevered prostheses. The overall failure rate was found to be 1% (95% CI: 0–3%), with low heterogeneity observed (I^2^ = 14.3%; *p* = 0.31). The results are illustrated in Figure 2.

#### 3.1.2. Failure of Cantilevered Implant-Supported Prostheses

Nine studies evaluated 226 ISPDCs. The overall failure rate was found to be 1% (95% CI: 0–3%), with no heterogeneity observed (I^2^ = 0%; *p* = 0.53). The results are presented in Figure 3.

#### 3.1.3. Complications Involving the Prosthesis and/or Prosthetic Abutment

Five studies investigated a total of 143 ISPDCs that reported complications such as ceramic chipping, screw loosening, and abutment fractures. The overall analysis revealed a pooled complication rate of 14% (95% confidence interval: 5–26%), with moderate heterogeneity (I^2^ = 68%; *p* = 0.01). A subgroup analysis was carried out to assess the influence of the number of supporting implants on the occurrence of complications related to ISPDCs. The pooled complication rate for prostheses supported by two implants was found to be 17% (95% CI: 5–37%). In contrast, the analysis of prostheses supported by a single implant (*n* = 119; total sample size) indicated a pooled complication rate of 14% (95% CI: 3–29%), with substantial heterogeneity (I^2^ = 76%; *p* < 0.01), as illustrated in Figure 4.

Other subgroup analysis was performed to evaluate the impact of the retention system on the rate of technical complications in ISPDCs. Four studies, which included a total of 108 ISPDCs, compared cement-retained prostheses with screw-retained prostheses. The analysis showed that cement-retained ISPDCs had a pooled complication rate of 12% (95% CI: 4–23%), with no observed heterogeneity (I^2^ = 0%; *p* = 0.3815). In contrast, screw-retained ISPDCs had a higher pooled complication rate of 20% (95% CI: 0–59%), along with significant heterogeneity (I^2^ = 89.5%; *p* = 0.0020). In summary, the analysis revealed a combined complication rate of 16% (95% CI: 4–32%), with substantial variability (I^2^ = 73.5%; *p* = 0.0101). These results are demonstrated in Figure 5.

#### 3.1.4. Retention Loss

A total of seven studies evaluated 177 ISPDCs, focusing on complications related to loss of retention, including screw loosening, decementation, and loss of retention. The global analysis, performed using a random-effects model with the inverse variance method, revealed a pooled complication rate of 13% (95% CI: 7–21%), with low to moderate heterogeneity (I^2^ = 34.8%; *p* = 0.16). These findings are presented in Figure 6.

A subgroup analysis based on retention type, involving 126 ISPDCs, revealed that cement-retained prostheses had a pooled complication rate of 16% (95% CI: 3–36%), with moderate heterogeneity (I^2^ = 69.9%; *p* = 0.0360). In comparison, screw-retained prostheses showed a lower complication rate of 10% (95% CI: 3–19%) and no observed heterogeneity (I^2^ = 0%; *p* = 0.3436). The overall meta-analysis indicated a pooled complication rate of 13% (95% CI: 5–24%), with moderate heterogeneity (I^2^ = 53.8%; *p* = 0.0703). These findings are illustrated in Figure 7.

#### 3.1.5. Technical Complications in Relation to Cantilever Length

A total of six studies examined 143 ISPDCs, focusing on various technical complications, including issues related to the prostheses or abutment and loss of retention. The pooled analysis revealed an overall complication rate of 29% (95% CI: 16–43%) with moderate heterogeneity (I^2^ = 66.4%; *p* = 0.01). In a subgroup analysis of ISPDCs with cantilever extensions greater than 7 mm, the pooled complication rate was 36% (95% CI: 8–71%), exhibiting high heterogeneity (I^2^ = 80.5%; *p* = 0.02). Conversely, for cantilever extensions shorter than 7 mm (n = 99 prostheses), the complication rate was 25% (95% CI: 13–39%), with moderate heterogeneity (I^2^ = 53.5%; *p* = 0.09). These findings are illustrated in Figure 8.

#### 3.1.6. Occlusal Veneering Material

Five studies evaluated 126 ISPDCs focusing on various technical complications such as issues related to the prostheses or abutment and loss of retention. These analyses examined the impact of different occlusal veneering materials, including metal, full zirconia, and porcelain-fused-to-metal (PFM). In the subgroup analysis, ISPDCs with metal occlusal surfaces exhibited a pooled complication rate of 17% (95% CI: 4–41%). ISPDCs made entirely of zirconia (n = 31) showed a slightly lower complication rate of 13% (95% CI: 3–30%). Conversely, PFM ISPDCs (n = 77) had a higher complication rate of 38% (95% CI: 20–57%), with moderate heterogeneity (I^2^ = 65.8%; *p* = 0.0538). The overall pooled analysis indicated a complication rate of 28% (95% CI: 14–45%), also displaying moderate heterogeneity (I^2^ = 72.9%; *p* = 0.0052). These findings are illustrated in Figure 9.

#### 3.1.7. Biological Complications

A total of six studies analyzed 175 implants, focusing on biological complications such as peri-implant mucositis, peri-implantitis, and gingival discomfort. The pooled analysis indicated a summary proportion of 20% (95% CI: 4–43%), with significant heterogeneity (I^2^ = 89.5%; *p* < 0.01). A subgroup analysis of ISPDCs supported by two implants reported a pooled rate of biological complications of 17% (95% CI: 3–63%). In contrast, an analysis of ISPDCs supported by a single implant showed a complication rate of 21% (95% CI: 4–46%), also with substantial heterogeneity (I^2^ = 91%; *p* < 0.01). However, there was no significant difference between the two groups analyzed (*p* = 0.9353). Further subgroup analysis based on study design (randomized controlled trials, prospective, and retrospective studies) revealed varying complication rates according to the type of study. For randomized controlled trials assessing implant-supported prostheses on a single implant, the complication rate was found to be 17% (95% CI: 5–42%). Prospectively designed studies revealed higher complication rates: 34% (95% CI: 0–94%) for prostheses supported by a single implant, with heterogeneity of I^2^ = 94.8%; *p* < 0.0001, and 17% (95% CI: 3–63%) for those supported by two implants. In contrast, retrospective studies evaluating cantilevers supported by a single implant reported a biological complication rate of 14% (95% CI: 0–42%), with heterogeneity of I^2^ = 89.9%; *p* < 0.0001, as illustrated in Figure 10.

#### 3.1.8. Marginal Bone Loss (Final Time)

Seven studies evaluated a total of 153 implants, focusing on marginal bone loss at the final follow-up. The pooled analysis indicated a mean marginal bone loss of 0.85 mm (95% CI: 0.40–1.31), with substantial heterogeneity (I^2^ = 98.5%; *p* < 0.01), as shown in Figure 11. A subgroup analysis based on study design as randomized controlled trials, prospective studies, and retrospective studies, revealed variations in marginal bone loss according to the study type. Randomized controlled trials reported a mean marginal bone loss of 1.50 mm (95% CI: 1.42–1.57), with low heterogeneity (I^2^ = 7.4%; *p* = 0.03396). Conversely, prospective studies showed a mean marginal bone loss of 0.96 mm (95% CI: 0.83–1.09). In contrast, retrospective studies reported a lower mean marginal bone loss of 0.29 mm (95% CI: 0.02–0.57), accompanied by a high level of heterogeneity (I^2^ = 81.9%; *p* = 0.0009). There was a significant difference among the study designs (*p* < 0.0001).

#### 3.1.9. Marginal Bone Loss Final Minus Initial Time (TF-T0)

A meta-analysis was conducted using data from four studies, including 89 implants at the final time point and 93 implants at the initial time. The analysis found that there was significantly greater marginal bone loss at the final time compared to the initial time, with a mean difference (MD) of 0.35 mm (95% confidence interval: 0.04–0.66; *p* < 0.0001). Additionally, substantial variability was observed in the results, as indicated by an I^2^ value of 82.3% (*p* = 0.0002). These findings are illustrated in Figure 12. According to the study design, an additional subgroup analysis revealed variability in marginal bone loss over time (final less baseline). For randomized controlled trials (RCTs), the pooled mean difference was 0.62 mm (95% CI: 0.44–0.80), with low heterogeneity (I^2^ = 12%; *p* = 0.3211). Prospective and retrospective studies showed minimal changes in marginal bone levels, with mean differences of 0.04 mm (95% CI: –0.16 to 0.24) and –0.04 mm (95% CI: –0.58 to 0.50), respectively. There was a significant difference between the time points (*p* < 0.0001).

#### 3.1.10. Marginal Bone Loss (Adjacent vs. Distant Sites)

Four studies evaluated a total of 109 implants, comparing marginal bone loss at sites adjacent to the cantilever with those on the opposite (distant) side. The pooled analysis showed a mean difference of 0.01 mm (95% CI: –0.15 to 0.16), with no observed heterogeneity (I^2^ = 0%; *p* = 0.78). In the subgroup analysis focused on single-implant-supported prostheses, the mean difference was 0.00 mm (95% CI: –0.21 to 0.21), which also showed no heterogeneity (I^2^ = 0%; *p* = 1.00). For cantilevered prostheses supported by two implants, the mean difference in bone loss between adjacent and distant sites was again 0.01 mm (95% CI: –0.21 to 0.23), with I^2^ = 0%; *p* = 0.42. These findings are illustrated in Figure 13. Overall, all analyses performed indicated no statistically significant differences (*p* = 0.93).

### 3.2. Risk and Bias Analysis

Out of the studies reviewed, only two were randomized controlled trials (RCTs) that showed a low risk of bias due to their adherence to appropriate methodological designs. Notably, only one of these studies calculated the sample size to establish the number of participants included [22] (see Figure 14).

Nine studies were non-randomized and utilized clinical follow-up designs, including case series, retrospective and prospective studies. Most of these studies received adequate scores on the Newcastle–Ottawa Scale (NOS), with scores greater than 6; however, one case series lacked a control group [28]. Several studies demonstrated additional methodological strengths, such as blinded outcome assessments conducted by independent dentists [26] and blinded evaluations of radiographic examinations [15]. In one study, randomization was specifically applied to assess implant-related marginal bone loss [2], while another included independent statistical analyses performed by a professional statistician [25]. Full methodological quality assessments can be found in Appendix A.

### 3.3. Assessing the Certainty of the Evidence

The overall quality of evidence for the primary outcomes was rated low according to the GRADE approach. Several factors influenced this assessment, including study design, inconsistencies, imprecision, and publication bias. It is noteworthy that only two of the included studies were randomized controlled trials (RCTs), and most studies did not adequately report on aspects such as consecutive sequence generation and allocation concealment. Moreover, some articles were either retrospective or consisted of case series. Additionally, many studies had small sample sizes or did not include sample size calculations, and a limited number of studies were available. Significant heterogeneity was observed (I^2^ > 40%), and non-overlapping confidence intervals were noted. The number of studies was deemed insufficient for meaningful comparison between the control and test groups for specific outcomes. While the data identified in this evaluation may seem promising, careful extrapolation is necessary. Further details can be found in Table 2 and Appendix A.

## 4. Discussion

The primary objective of this systematic review was to assess the survival rates and associated complications of implant-supported partial dentures with cantilever extensions (ISPDCs) in the posterior region. The results show that although ISPDCs experience more technical complications, especially with loss of retention and chipping of the occlusal material, these issues do not affect their longevity or survival rates.

In the analyzed sample, a prosthesis and implant failure rate of 1% (95%IC: 0–3%). This rate aligns with findings from existing literature on implants supporting non-cantilevered prostheses (ISNCP) [31,32,33]. For example, Zurdo et al. [31] reported a survival rate of 96.3% to 96.2% for implant-supported prostheses without cantilever extensions, with a weighted mean survival rate of 95.8%. Additionally, other systematic reviews have indicated that cantilever extensions do not negatively impact the survival of implant-supported prostheses when compared to ISNCP [32,33].

The incidence of complications related to prostheses or their abutments, such as ceramic chipping, screw fractures, and abutment damage, was observed in 14% of cases. When analyzing these technical complications based on the type of prosthesis retention (screwed versus cemented), screwed prostheses exhibited 8% higher complication rate than cemented ones. However, our results did not show a significant difference between the two groups, which is consistent with findings from a previous systematic review that also found no significant difference in complications between cemented and screwed implant-supported prostheses [34].

A retention loss rate of 13% was observed, which included cases of screw loosening and decementation. In the subgroup analysis comparing retention types (screwed versus cemented), it was found that cemented prostheses had 6% higher rate of retention loss. Another systematic review focused on implant-supported fixed prostheses indicated that technical complications, such as ceramic chipping, were significantly more common in screw-retained rehabilitations compared to those that were cemented. Conversely, abutment loosening occurred more frequently in cemented prostheses [35].

There is currently no consensus in the literature regarding the most appropriate retention method for cantilevered prostheses—screw-retained versus cemented. Some authors suggest that screw-retained prostheses make managing potential complications easier [29,36]. In contrast, others argue that cemented prostheses provide more favorable stress distribution from a biomechanical perspective [30]. Additionally, there is uncertainty about the optimal implant connection type—internal or external—for implant-supported prosthetic dental crowns. The present sample does not provide enough data to support specific recommendations. Therefore, further well-designed clinical studies are needed to investigate this issue.

When discussing the complications associated with implant-supported fixed partial dentures, the type of material used in the prosthesis plays a crucial role. Although only a few studies in this review report overall complication rates based on occlusal material, those that do suggest that prostheses fabricated from metal or zirconia tend to exhibit lower complication rates compared to those made from metal ceramic. Galal et al. [30] emphasize the importance of minimizing cuspal inclinations when using metal prostheses to reduce masticatory overload and enhance vertical forces. Similarly, Roccuzzo et al. [15] analyzed zirconia prostheses and highlighted the need for careful occlusal control to prevent premature contacts in cantilever extensions. These recommendations are applicable regardless of the occlusal material used. Given the diverse mechanical properties and clinical behaviors of materials used in ISPDCs, we strongly recommend that future studies specifically investigate the performance of these materials, particularly with the recent introduction of new monolithic materials available in the market.

From a biomechanical perspective, the length of a cantilever is directly related to the amount of stress transferred to prosthetic components and around surrounding bone. A previous clinical study found that implant-supported partial dentures without complications, after an average follow-up period of 51 months, had a mean cantilever length of less than 8 mm. Based on this finding, the authors recommend a maximum cantilever length of 7 mm for ISPDCs [37]. Our review showed that studies examining prostheses with cantilever lengths exceeding 7 mm reported an 11% higher complication rate compared to those with lengths shorter than 7 mm. This increase includes issues such as retention loss and complications related to the prosthesis and abutments. These results emphasize the importance of careful and precise planning when designing ISPDCs concerning cantilever length [38]. However, due to the limited data available on this topic, it is advisable to conduct new clinical studies focusing on this aspect of ISPDCs.

Although the observed technical complication rates are higher than the 5.1% to 9.7% typically reported in the literature for partial dentures without cantilever extensions, the complications observed in ISPDCs are generally considered minor. They can be managed effectively in routine clinical practice [31,32]. These complications may be attributed to the increased stress concentration experienced by ISPDCs, especially under non-axial loading conditions, compared to partial dentures without cantilever extensions [6,7,8]. Such loading generates tension on the metal components and prosthetic screws, and the repetitive nature of these forces disrupts the mechanical interlocking of the components, ultimately leading to loss of retention [29]. Furthermore, non-axial forces are known to induce shear stresses that compromise the internal structural cohesion of prosthetic materials. This mechanical disruption may contribute to ceramic chipping, fracture of prosthetic components, and decementation of the restorations [29,38].

The findings reinforce the recommendation for designing occlusal cusps with reduced inclination and height [30], ensuring precise occlusal adjustments to eliminate contact during excursive movements [38], and implementing strict control protocols in patients with bruxism, particularly in the posterior region where masticatory forces are more intense. When these clinical criteria are followed, the risk of prosthesis failure is significantly minimized.

From a biomechanical perspective, the forces exerted on implant-supported partial dentures can exceed the structural limits of narrow-diameter implants (less than 3.5 mm), leading to potential fractures. In a study by Hälg et al. [2], two cases of implant fracture were reported, emphasizing the contraindication for using narrow-diameter implants in rehabilitation involving ISPDCs. Additionally, a study by Thoma et al. [22] indicated that using short implants (less than 8 mm) to support ISPDCs resulted in significantly higher rates of biological complications. This raises concerns about the reliability of short implants in ISPDCs, highlighting the need for further clinical research to explore this issue.

The overall rate of biological complications observed was 20%, which includes cases of mucositis and peri-implantitis. This rate is significantly higher than the 5.7% to 9.6% reported for implants supporting non-cantilever prostheses [39,40]. An analysis of the types of studies conducted showed that prospective studies and randomized controlled trials (RCTs) had higher rates of biological complications compared to retrospective studies. This suggests that more rigorous monitoring in prospective studies and RCTs provides more reliable clinical data [15,28,30]. The high rates of biological complications emphasize the importance of consistent periodontal monitoring for patients rehabilitated with ISPDCs.

In particular, given the increased difficulty of cleaning in these cases, the importance of prosthetic designs that facilitate adequate access for cleaning beneath the prosthesis is emphasized, thereby reducing the risk of inflammation and marginal bone loss. When analyzing the cumulative bone loss in ISPDCs supported by a single implant, the results fit the expected clinical standards for prostheses without cantilever extensions in the same region. Furthermore, although several in vitro and silico studies have demonstrated differences in biomechanical behavior and stress distribution between areas adjacent and distant from the cantilever [41,42,43], the present analysis did not find a clinically significant difference in the mean marginal bone loss between these regions. Overall, regarding marginal bone loss, the present study’s findings support the clinical success of ISPDCs in the posterior region.

Biomechanical studies have shown differences in the behavior of the mesial and distal cantilevers [9,10]. However, clinical studies evaluating this aspect have found no significant differences in the clinical performance of ISPDCs [3,24]. Additionally, when comparing prostheses supported by one versus two implants, previous studies indicate that using two or more implants results in better stress distribution than those supported by a single implant [44]. Nevertheless, further clinical studies are needed to establish evidence-based recommendations.

The GRADE system assesses the certainty of evidence, emphasizing the need for prospective controlled studies and randomized controlled clinical trials to investigate the analyzed variables. Additionally, it is important to conduct sample size calculations for patient compositions. When these elements are considered alongside the diversity of study designs, they may contribute to increased heterogeneity and non-overlapping confidence intervals. This highlights the necessity for a careful evaluation of the currently available information regarding ISPDCs.

Among the limitations of this review, it is important to note the inclusion of heterogeneous study designs (prospective, retrospective, and randomized clinical trials), which were analyzed collectively and may have contributed to increased data heterogeneity. Additionally, several studies presented missing data, particularly regarding the types of implants used, cantilever length, and occlusal veneering material, representing a relevant limitation. Furthermore, the limited number of studies with control groups represents a constraint, partly due to the exclusion of articles that evaluated ISPDCs in the posterior region (Appendix A). Future clinical studies should consider stratifying key variables to improve the quality of evidence and support clinical decision-making. These should include prosthesis location (anterior or posterior), type of retention (screw-retained or cemented), occlusal material, length of cantilever extension, and the number of supporting implants per prosthesis.

## 5. Conclusions

Based on the results obtained, it was possible to conclude that
The failure rate of both prostheses and implants was low, at 1% among the studies evaluated. However, the incidence of technical complications including prosthesis and abutment-related issues, as well as loss of retention, was notable, affecting up to 29% of cases, particularly in cases involving cantilever extensions greater than 7 mm. These findings emphasize the importance of meticulous treatment planning and continuous clinical monitoring in such rehabilitations.Biological complications such as peri-implant mucositis and peri-implantitis were notable, occurring in 20% of cases, indicating a need for periodontal monitoring.Significant differences were observed in marginal bone loss between the baseline and final time points, although the values remained clinically acceptable. The study design appears to influence marginal bone loss, indicating the need for further investigation. Additionally, the location of implant sites, whether adjacent to or distant from the cantilever, did not affect marginal bone loss. Overall, these results suggest favorable survival rates for ISPDCs in the posterior region.There was no significant influence observed from the type of occlusal veneering material, the number of implants, or the retention system in the ISPDCs.Due to the limitations identified in the available studies and the potential heterogeneity in study designs, further well-designed clinical trials are necessary to support the development of evidence-based guidelines regarding prosthetic materials, the use of reduced-diameter or short implants, retention methods, and cantilever dimensions.Although the findings were generally favorable, the overall certainty of the evidence was rated as low due to methodological limitations and substantial heterogeneity, highlighting the need for cautious interpretation and the conduction of randomized controlled trials in future research.


## Figures and Tables

**Figure 1 materials-18-04704-f001:**
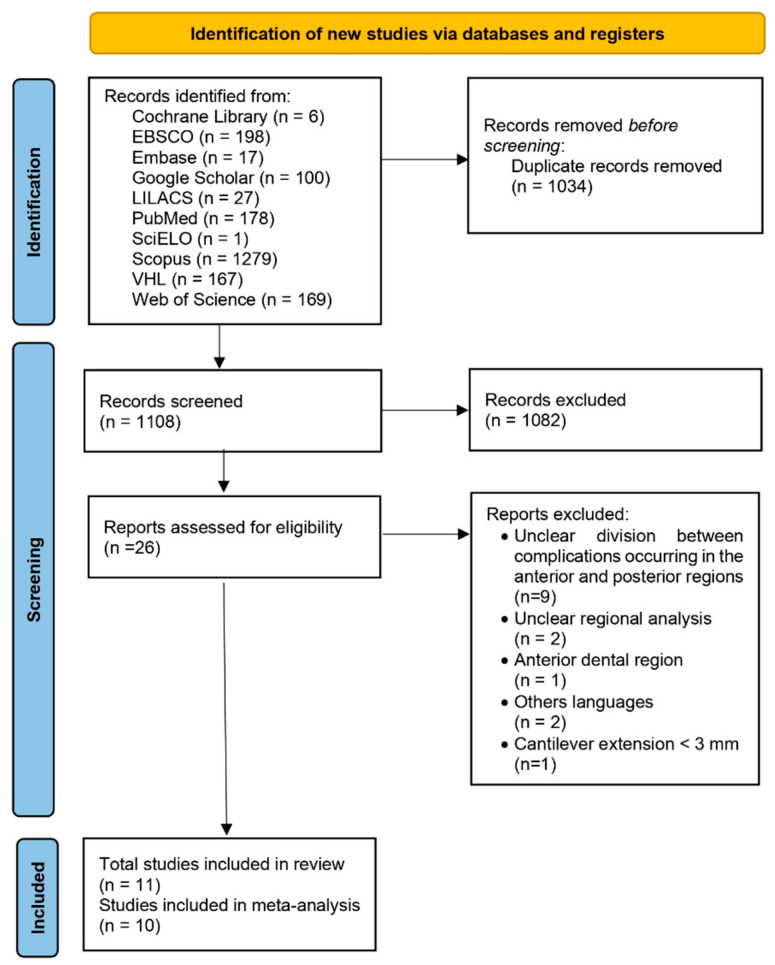
PRISMA flow diagram showing each step of the study selection process for the systematic review, from the initial database identification to the final inclusion after applying eligibility criteria.

**Figure 2 materials-18-04704-f002:**
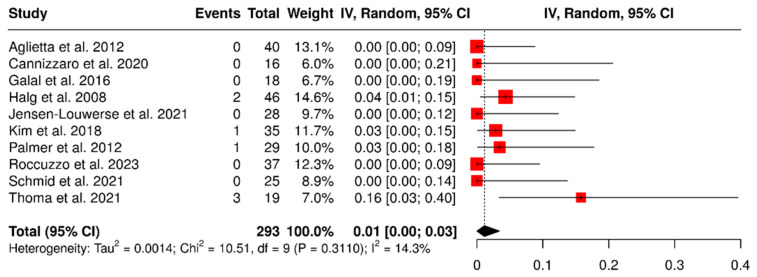
Forest plot showing failure rates of implants that support partial dentures with cantilever extensions, based on data from studies [2,3,15,22,24,26,27,28,29,30].

**Figure 3 materials-18-04704-f003:**
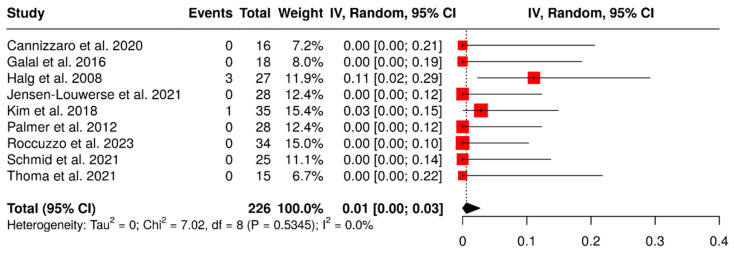
Forest plot illustrating failure rates of implant-supported partial dentures with cantilever extensions, derived from data across nine studies evaluating this variable [2,3,15,22,26,27,28,29,30].

**Figure 4 materials-18-04704-f004:**
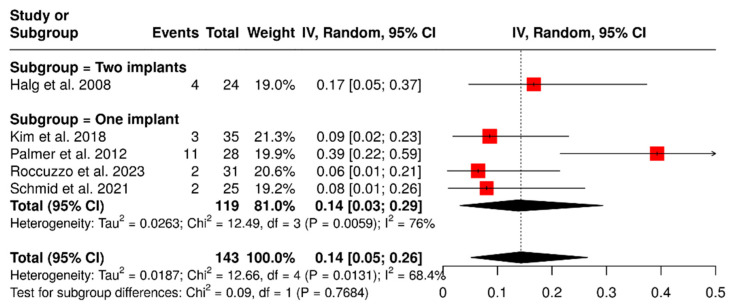
Forest plot illustrating complications related to prosthetics and abutments, including issues such as ceramic chipping, screw loosening, and abutment fractures in implant-supported partial dentures with cantilever extensions, categorized by the number of supporting implants (single implant vs. two implants) [2,15,26,27,29].

**Figure 5 materials-18-04704-f005:**
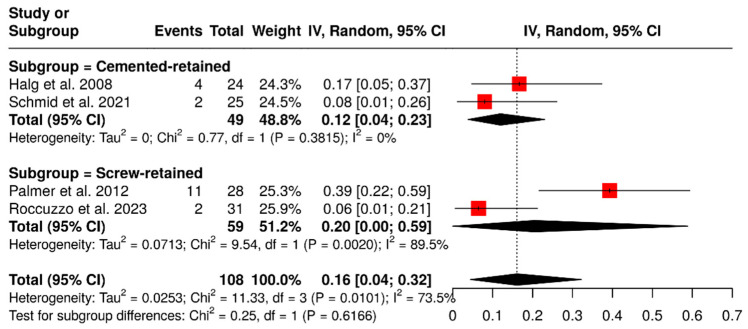
Forest plot illustrating the technical complication rates of implant-supported partial dentures with cantilever extensions, comparing screw-retained and cement-retained prostheses [2,15,27,29].

**Figure 6 materials-18-04704-f006:**
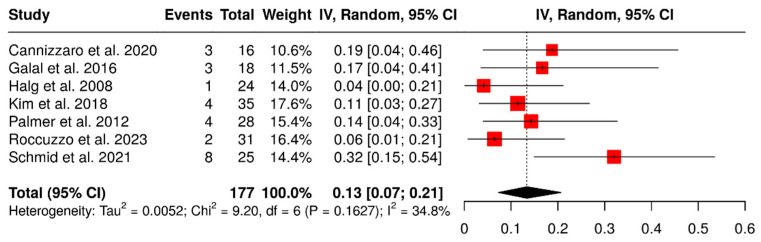
Forest plot illustrating retention-related complications: screw loosening, decementation, and loss of retention, in implant-supported partial dentures with cantilever extensions, based on data from seven studies [2,15,26,27,28,29,30].

**Figure 7 materials-18-04704-f007:**
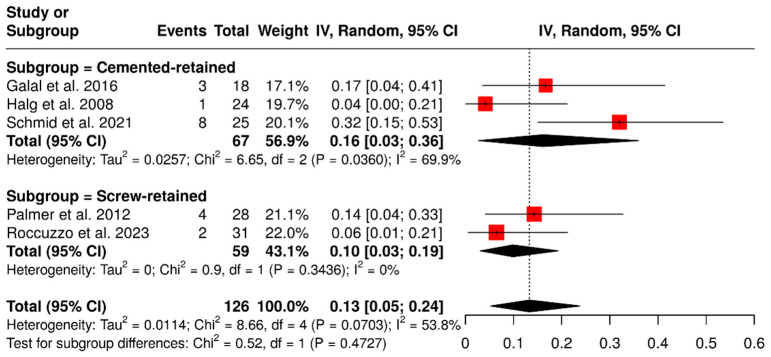
Forest plot illustrating retention-related complications (screw loosening, decementation, and loss of retention) in implant-supported partial dentures with cantilever extensions, comparing cement-retained and screw-retained prostheses [2,15,27,29,30].

**Figure 8 materials-18-04704-f008:**
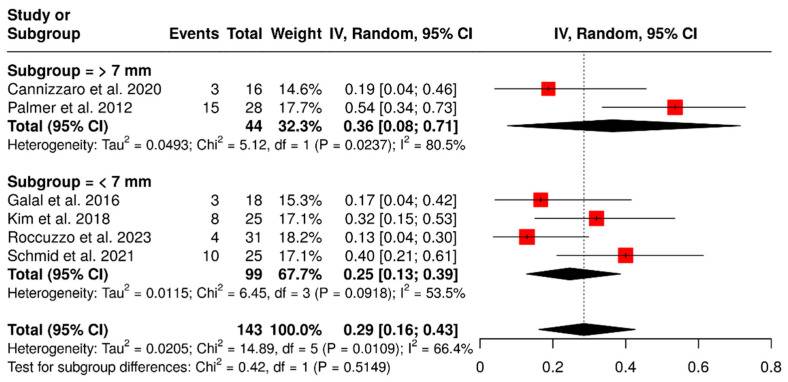
Forest plot illustrating technical complications in implant-supported partial dentures with cantilever extensions, categorized by cantilever length (>7 mm vs. <7 mm) [15,26,27,28,29,30].

**Figure 9 materials-18-04704-f009:**
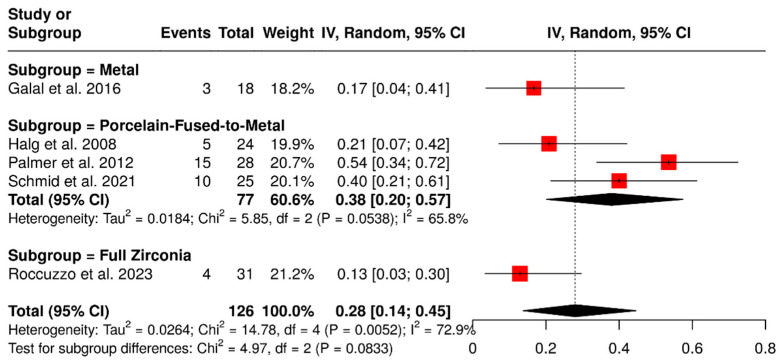
Forest plot of technical complications, including loss of retention and issues with prostheses and abutments, in implant-supported partial dentures with cantilever extensions, categorized by the type of occlusal veneering material [2,15,27,29,30].

**Figure 10 materials-18-04704-f010:**
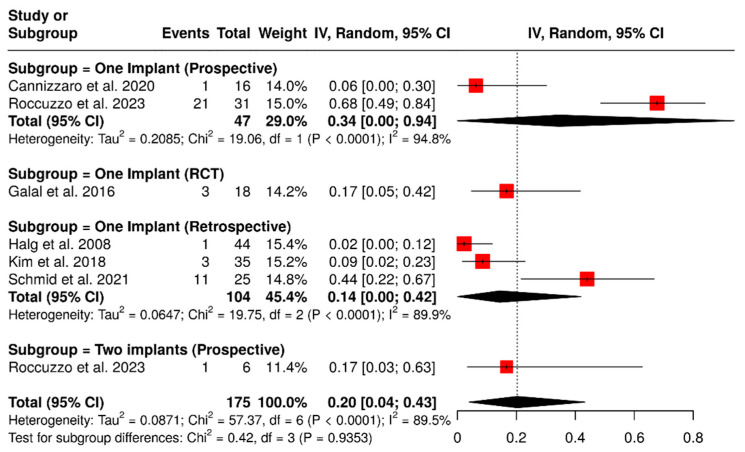
Forest plot of subgroup analysis showing biological complication rates in implant-supported partial dentures with cantilever extensions, categorized by study design (RCTs, prospective, and retrospective) and the number of supporting implants (one or two) [2,15,26,27,28,30].

**Figure 11 materials-18-04704-f011:**
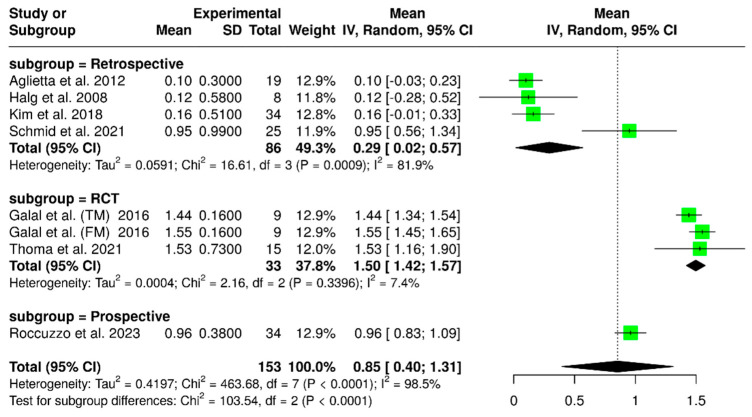
Forest plot analyzing mean marginal bone loss in implant-supported partial dentures with cantilever extensions, categorized by study design (RCTs, retrospective, prospective) [2,15,22,24,26,27,30].

**Figure 12 materials-18-04704-f012:**
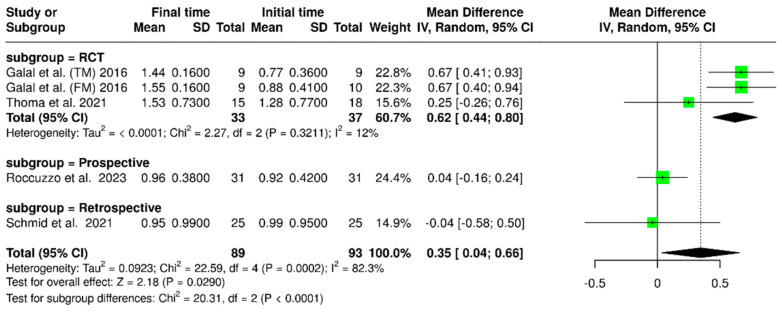
Forest plot of marginal bone loss comparing baseline and final time points. [15,22,27,30].

**Figure 13 materials-18-04704-f013:**
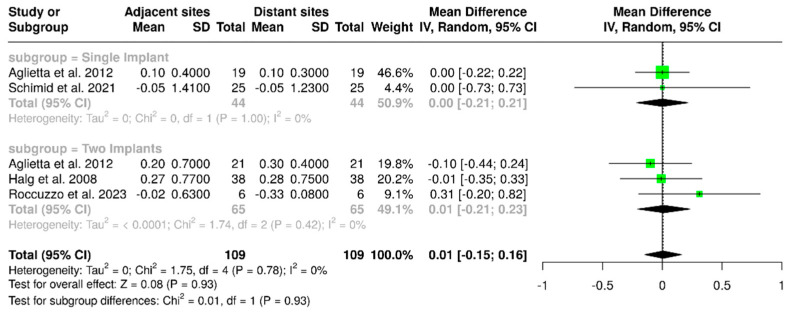
Forest plot comparing marginal bone loss between implant surfaces that are adjacent to and distant from the cantilever in implant-supported partial dentures with cantilever extensions, along with subgroup analyses based on the number of implants (one or two implants) [2,15,24,27].

**Figure 14 materials-18-04704-f014:**
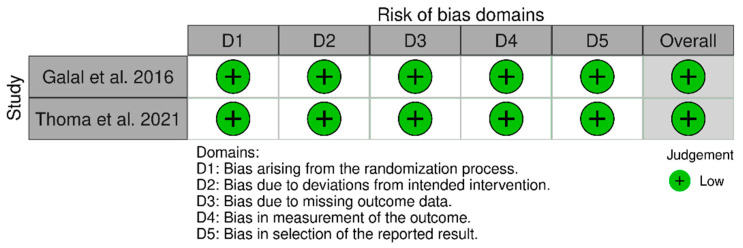
Assessment of risk of bias included randomized controlled trials using the ROB scale [22,30].

**Table 1 materials-18-04704-t001:** Data extracted from selected studies.

Author and Year	Desing	Follow-Up (M)	Average Age (Min-Max)	N° Patients	N° Impl.	Length × Diameter of Impl.	Max/ Man.	N° Prost. Total/ Cant.	Super-Structure	Cant. Length (mm)	Mesial/Distal Cant.	Screwed or Cemented	Survival Impl./Prost.	Technical/ Biological Complications
Aglietta et al. 2012 [24]	Retros. (III-2)	67.8–78.2	G1: 54.6 ± 12.9 G2: 60.5 ± 7.7	38	61	NR × 4.1–4.8 Ø	19/21	40/40	CM	NR	18/22	Both	100%	NR
Cannizzaro et al. 2020 [28]	Prosp. Case series (III-3)	24	62.8 (44–76)	16	16	8.5–10 × 5 Ø	16 max.	16/16	MC and Fiberglass	10–12	16 distal	Both	100%	3/1
Dereci et al. 2021 [25]	Retros. (III-2)	24	57.9 ± 8.3	52	104	NR	104 max.	52/22	NR	NR	22 distal	NR	NR	NR
Galal et al. 2016 [30]	RCT (II)	36	41.1 ± 11	18	18	10 × 4.5 Ø	18 mand.	18/18	Ni-Cr	3 and 5	18 distal	Cem.	100%	3/3
Hälg et al. 2008 [2]	Retros. (III-2)	63.6	24–83	54 (T0), 49 (FT)	78	6–12 × 3.3–4.8 Ø	31/47	54/27	Au + C	NR	12/15	Cem.	CG-96.9/96.3%. Cant.-95.7/88.9%	5/1 Frac. 1 prot. and frac. 2 impl.
Jensen Louwerse et al. 2021 [3]	Retros. Case series (III-3)	78	64 (19–84)	23	28	8–13 × 4 Ø	15/13	28/28	CM e Zir	2 teeth	23/5	Both	100%	*
Kim et al. 2018 [26]	Retros. (III-3)	47.72	57.8 (26–71)	33	35	7–13 × 4.3–7 Ø	17/18	35/35	Nr	3.01–5.99	33/2	NR	100%	7/3 Frac. 1 prot. and 1 impl. lost
Palmer et al. 2012 [29]	Prosp. (III-2)	36	50 (18–70)	29	28	9–15 × 4–5 Ø	20/8	28/28	CM	8	4/24	Scre.	96.6/100%	11/NR
Roccuzzo et al. 2023 [15]	Prosp. (III-2)	31.1	67.7 ± 9.2	35 (T0), 30 (FT)	37	8–12 × 4.1–4.8 Ø	30/7	34/34	Zir	6.3	24/10	Scre.	100%	4/22
Schmid et al. 2021 [27]	Retros. (III-3)	163.2	71.4 (45–89)	21	25	8–12 × 4.1–4.8 Ø	10/15	25/25	CM	5.5	16/9	Cem.	100%	10/11
Thoma et al. 2021 [22]	RCT (II)	60	67.5 ± 11.6	36	54	6 × 4.1 Ø	54 max. and mand.	36/18	CM	NR	18 mesial (ideally)	Scre.	CG-80.4%./NR Cant.-84.2%/NR	18/*

The symbol Ø denotes the diameter of the implant. Au = gold; C = ceramic; Cant. = cantilever; Cem. = cemented; CG = control group; CM = ceramo-metal; Frac. = fracture; FT: Final time; Impl. = implants; M = months; Man = mandible; Max = maxilla; Ni-Cr = nickel-chromium; MC: Metal-Composite; NR = not reported; Prosp. = prospective; Prost. = prosthesis; RCT = randomized controlled trials; Retros. = retrospective; Scre. = screwed; T0: Initial time; Zir = zircônia; * = studies that presented data in percentages. For example, Jensen-Louwerse et al. 2021 [3] reported the following biological complications: peri-implant mucositis at 89.3% and peri-implantitis at 17.9%. The technical complications included veneering ceramic fracture at 7.1%, screw loosening at 3.6%, and decementation at 3.6%. Thoma et al. 2021 [22] reported peri-implant mucositis rates of 56.2% in the cantilever group and 63.6% in the two-implants group.

**Table 2 materials-18-04704-t002:** GRADE Summary of Findings for Main Clinical Outcomes.

Outcome	No. of Studies	N° of Participants (Studies)	Certainty of Evidence	Reasons for Downgrading
Failure of implants in cantilever prostheses	10	230	⨁◯◯◯ Very low	a, b
Failure of implant-supported prostheses with cantilever	9	209	⨁◯◯◯ Very low	a, b
Complications involving the prosthesis and/or abutment	5	137	⨁◯◯◯ Very low	a, b, c, d, e
Prosthetic retention loss	7	171	⨁◯◯◯ Very low	a, b
Technical complications related to cantilever length	6	147	⨁◯◯◯ Very low	a, c, d, e
Biological complications	6	142	⨁◯◯◯ Very low	a, c, d, e
Marginal bone loss (final timepoint) stratified by study design	7	153	⨁◯◯◯ Very low	a, b, c
Change in marginal bone loss (baseline to final) stratified by study design	4	93	⨁◯◯◯ Very low	a, b, c
Marginal bone loss at adjacent vs. distant implant sites	4	109	⨁◯◯◯ Very low	a, b
Complication rates comparing screw-retained and cement-retained prostheses	4	129	⨁◯◯◯ Very low	a, b, c
Retention rates comparing screw-retained and cement-retained prostheses	5	147	⨁◯◯◯ Very low	a, b, c, e
Technical complications in ISPDCs, according to the type of occlusal veneering material	5	147	⨁◯◯◯ Very low	a, b, c

a Random sequence generation and allocation concealment; b Small sample size or absence of sample size calculation; c There is significant heterogeneity, with I^2^ values greater than 40%; d non-overlapping confidence intervals; e The number of studies for this outcome has been reduced.

## Data Availability

The original contributions presented in this study are included in the article/Appendix A. Further inquiries can be directed to the corresponding author.

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
