# Peer review of "Implant-Supported Cantilever Fixed Partial Dentures in the Posterior Region: A Systematic Review and Meta-Analysis on Survival Outcomes"

_materials, 2025, doi:10.3390/ma18204704_

Round 1
Reviewer 1 Report
Comments and Suggestions for Authors
This systematic review and meta-analysis is designed to assess the survival, complications, and marginal bone loss related to posterior implant-supported partial dentures with cantilever extensions. The implant and prosthesis failure rates were low (1%), and the evidence suggests that the complications are associated with cantilever extensions beyond 7 mm. The study is methodologically well-executed, followed PRISMA guidelines, and provided relevant clinical insights into implant-supported prosthodontics. This study addressed a gap in focused evidence-based evaluation, as the implant-supported cantilever prostheses in the posterior region are of increasing clinical relevance.
General Comments
- The review is original, focusing on one area (posterior region) and providing clinical implications that were not covered in previous reviews.
- Most of the references are from the last five years and relevant to the topic.
- The inclusion and exclusion are criteria GRADE uses for quality assessment, and the database search is comprehensive.
Specific Comments
- Figures 6 & 7 are necessary to understand the complication patterns. I suggest integrating more explanation of these figures into the discussion.
- Table 1: Included several "NR" (not reported) values for important variables (cantilever length and prosthesis material)!
- Abbreviation: Define less common abbreviations such as “FT”, “T0” where they are first used. Also, ensure consistent use of ISPDCs throughout the text.
Language and Grammar: The manuscript is well-written. However, some grammatical improvements are suggested:
Line 28: Replace the word “need” with “require” in the statement “...a need for more cautious case selection...”
Line 337: Replace the word “can” with “may” in the statement “...can contribute to ceramic chipping...”
- The definition of "long" cantilevers varies in literature. This study reported that technical complications are more common in cantilever extensions >7 mm, which is important. Please clarify whether 7 mm is based on literature consensus or a biomechanical threshold.
- I think that missing cantilever length in five studies is a limitation. So, it should be highlighted in the discussion.
- I suggest taking into consideration cantilever direction, retention type, and the material used.
- Kindly emphasize more clearly in the abstract and conclusion the overall certainty of evidence rate" using GRADE. This will help to guide readers interpretation.
- Kindly double-check if there are outlier studies influencing the heterogeneity of marginal bone loss measurements.
Author Response
REVIEWER 1:
This systematic review and meta-analysis is designed to assess the survival, complications, and marginal bone loss related to posterior implant-supported partial dentures with cantilever extensions. The implant and prosthesis failure rates were low (1%), and the evidence suggests that the complications are associated with cantilever extensions beyond 7 mm. The study is methodologically well-executed, followed PRISMA guidelines, and provided relevant clinical insights into implant-supported prosthodontics. This study addressed a gap in focused evidence-based evaluation, as the implant-supported cantilever prostheses in the posterior region are of increasing clinical relevance.
General Comments
- The review is original, focusing on one area (posterior region) and providing clinical implications that were not covered in previous reviews.
- Most of the references are from the last five years and relevant to the topic.
- The inclusion and exclusion are criteria GRADE uses for quality assessment, and the database search is comprehensive.
R.: We thank Reviewer 1 for the positive and encouraging feedback regarding the clinical relevance and methodological rigor of our study. We are also grateful for the thoughtful considerations and constructive suggestions, which helped us to improve the clarity and overall quality of our manuscript. All suggestions and observations have been carefully addressed, and the corresponding changes in the manuscript are highlighted in yellow for easy identification.
Specific Comments
- Figures 6 & 7 are necessary to understand the complication patterns. I suggest integrating more explanation of these figures into the discussion.
R.: We appreciate your suggestions regarding Figures 6 and 7, particularly concerning the effect of cantilever length (<7 mm vs. >7 mm) and biological complications. The discussion related to these figures has been expanded, including a revision of Figure 7, which now presents a meta-analysis of biological complications stratified by study design (prospective, retrospective, or RCTs) to evaluate their respective effects on complication rates.
- Table 1: Included several "NR" (not reported) values for important variables (cantilever length and prosthesis material)!
R.: Table 1: We thank the reviewer for this observation. These data would indeed be important for meta-analyses. We attempted to contact the authors of the included studies (e.g., Roccuzzo et al., 2023) to supplement missing information. However, due to the unavailability of such data, we acknowledged this as a limitation of our study and emphasized the need for future clinical trials to address this gap.
- Abbreviation: Define less common abbreviations such as “FT”, “T0” where they are first used. Also, ensure consistent use of ISPDCs throughout the text.
R.: Due to space constraints in the tables, all abbreviations are fully defined in the corresponding captions, including those for initial and final timepoints. We also conducted a full-text review to ensure consistent usage of the acronym ISPDCs.
Language and Grammar: The manuscript is well-written. However, some grammatical improvements are suggested:
Line 28: Replace the word “need” with “require” in the statement “...a need for more cautious case selection...”
Line 337: Replace the word “can” with “may” in the statement “...can contribute to ceramic chipping...”
R.: We thank you for these suggestions; both terms have been revised accordingly in the manuscript.
- The definition of "long" cantilevers varies in literature. This study reported that technical complications are more common in cantilever extensions >7 mm, which is important. Please clarify whether 7 mm is based on literature consensus or a biomechanical threshold.
R.: Due to discrepancies in the literature regarding what constitutes a "long" cantilever, we followed the findings of Kim et al. (2014), which indicated that cantilevers shorter than 8 mm were associated with fewer complications. This threshold guided our subgroup analysis and has been further clarified in the discussion section.
References:
Kim, P; Ivanovski, S; Latcham, N.; Mattheos, N. The impact of cantilevers on biological and technical success outcomes of implant-supported fixed partial dentures. A retrospective cohort study. Clin Oral Implants Res. 2014, 25, 175-84. https://doi.org/10.1111/clr.12102.
- I think that missing cantilever length in five studies is a limitation. So, it should be highlighted in the discussion.
R.: The discussion on the relationship between complication rates and cantilever length has been enhanced, and this limitation is now clearly described in the limitations paragraph.
- I suggest taking into consideration cantilever direction, retention type, and the material used.
R.: As complication numbers were often presented as a global count without distinctions for cantilever direction, retention type, or occlusal material, data were insufficient for robust subgroup meta-analysis. However, we added a meta-analysis on prosthetic complications stratified by retention type (screw-retained/cemented), as well as one for global complications (including retention loss and prosthetic/abutment issues) by occlusal material. We noted the limited number of studies and advised cautious interpretation. Regarding cantilever direction (mesial/distal), the reviewed literature showed no significant influence, and we addressed this point in the discussion.
- Kindly emphasize more clearly in the abstract and conclusion the overall certainty of evidence rate" using GRADE. This will help to guide readers interpretation.
R.: Thank you for your valuable suggestion. We have now explicitly emphasized the overall certainty of the evidence, as assessed using the GRADE approach, in both the abstract and conclusion sections to better guide the reader's interpretation of our findings.
- Abstract: “The certainty of evidence was rated as low for all main outcomes. Study limitations, heterogeneity, and risk of bias influenced this assessment.”
- Conclusion: “Although the findings were generally favorable, the overall certainty of the evidence was rated as low due to methodological limitations and substantial heterogeneity, highlighting the need for cautious interpretation and the conduction of randomized controlled trials in future research.”
- Kindly double-check if there are outlier studies influencing the heterogeneity of marginal bone loss measurements.
R.: We appreciate your relevant comment. In response to your suggestion, we conducted subgroup analyses based on study design (randomized clinical trials, prospective cohorts, and retrospective studies), which contributed to explaining the observed heterogeneity. Randomized clinical trials showed a higher mean marginal bone loss, with values of 0.62 mm (95% CI: 0.44–0.80; I² = 12%) for the change between final and baseline time points (TF–T0), and 1.50 mm (95% CI: 1.42–1.57; I² = 7.4%; p = 0.03396) for marginal bone loss at the final follow-up. These outcomes, when evaluated separately, demonstrated lower heterogeneity among randomized controlled trials.
Thus, we conclude that, although heterogeneity persists in some cases, it is likely attributable to methodological differences among the included studies, being more pronounced in retrospective studies. This explanation has been incorporated into the revised version of the manuscript (see Results section, subsections 3.1.7 and 3.1.8).
Reviewer 2 Report
Comments and Suggestions for Authors
Dear authors,
This manuscript addresses an important clinical question regarding survival and complication rates of implant-supported partial dentures with cantilever extensions (ISPDCs) in the posterior region. The topic aligns with Materials (MDPI) scope, given the biomechanical and material considerations inherent in cantilever design and prosthetic fabrication.
Strengths:
Novelty and Scope: It focuses on the posterior region, where biomechanical challenges differ from anterior placements.
The protocol is registered in PROSPERO; adherence to PRISMA guidelines ensures transparency.
It includes ten major databases up to January 28, 2025.
The paper offers a meta-analysis for implant and prosthesis survival, technical and biological complication rates, and marginal bone loss.
It provides clear recommendations regarding cantilever length and clinical monitoring.
Weaknesses and Recommendations:
Heterogeneity and Study Design:
- The inclusion of retrospective case series, prospective cohorts, and only two RCTs contributes to substantial heterogeneity (I² values frequently >75%).
- Recommendation: Subgroup analyses by study design or sensitivity analyses excluding high-risk studies would strengthen conclusions.
Material-specific Performance:
- Mixed prosthetic materials were reported, but complications were not stratified by material type.
- You have to analyze outcomes by prosthetic material (e.g., metal–ceramic vs. zirconia) or, if data are insufficient, explicitly acknowledge this gap and suggest future studies.
Retention Method Discussion:
- Unclear data on cemented vs. screw-retained outcomes; two studies did not report retention type.
- Concerns raised regarding short (<8 mm) and narrow-diameter implants (<3.5 mm) supporting cantilevers.
- You have to highlight the need for standardized definitions and separate analyses for short/narrow implants; consider recommending caution in discussion.
GRADE Assessment Details:
- Evidence rated low quality, but summary of GRADE judgments is located only in the Supplementary Material. Include an abridged GRADE summary table in the main manuscript for reader clarity.
Minor Comments:
- Define all abbreviations at first use (e.g., ISPDCs).
- Provide exact p-values rather than ‘p > 0.05’ when possible.
- In the Introduction, streamline background citations to avoid redundancy.
This manuscript offers valuable insights into ISPDC performance in the posterior region. Addressing heterogeneity through additional analyses, stratifying by material and retention method, and enhancing clarity will significantly improve the manuscript’s impact and suitability for publication in Materials (MDPI).

Language and Clarity:
- Several typographical inconsistencies (e.g., missing spaces, superfluous line breaks).
- Thorough language editing to improve readability and ensure compliance with MDPI formatting, or get the help of a Native speaker.
Author Response
REVIEWER II –
Dear authors,
This manuscript addresses an important clinical question regarding survival and complication rates of implant-supported partial dentures with cantilever extensions (ISPDCs) in the posterior region. The topic aligns with Materials (MDPI) scope, given the biomechanical and material considerations inherent in cantilever design and prosthetic fabrication.
Strengths:
Novelty and Scope: It focuses on the posterior region, where biomechanical challenges differ from anterior placements.
The protocol is registered in PROSPERO; adherence to PRISMA guidelines ensures transparency.
It includes ten major databases up to January 28, 2025.
The paper offers a meta-analysis for implant and prosthesis survival, technical and biological complication rates, and marginal bone loss.
It provides clear recommendations regarding cantilever length and clinical monitoring.
Weaknesses and Recommendations:
R.: We sincerely thank Reviewer 2 for the thoughtful and detailed feedback, as well as for recognizing the relevance and strengths of our study. All comments and recommendations have been carefully considered. We performed additional subgroup analyses as suggested to address heterogeneity, and the corresponding modifications in the manuscript are highlighted in green for easy identification.
Heterogeneity and Study Design:
- The inclusion of retrospective case series, prospective cohorts, and only two RCTs contributes to substantial heterogeneity (I² values frequently >75%).
- Recommendation: Subgroup analyses by study design or sensitivity analyses excluding high-risk studies would strengthen conclusions.
R.: Thank you for your insightful comment. We fully agree with your observation regarding the potential influence of study design on heterogeneity. In response, we conducted additional meta-analyses with subgroup analyses based on clinical study design (randomized controlled trials, prospective, and retrospective studies) in outcomes where high heterogeneity was observed (I² > 75%). These results are now presented in the revised manuscript and discussed accordingly to improve the clarity and robustness of our conclusions.
Material-specific Performance:
- Mixed prosthetic materials were reported, but complications were not stratified by material type.
- You have to analyze outcomes by prosthetic material (e.g., metal–ceramic vs. zirconia) or, if data are insufficient, explicitly acknowledge this gap and suggest future studies.
R: As the complication data in the included studies were generally reported as a global figure without specification of prosthetic material types, we acknowledge the difficulty in conducting a detailed material-based subgroup analysis. Nonetheless, we have included a new meta-analysis on global complications (including retention loss and prosthetic/abutment complications) stratified by occlusal material used. Due to the limited number of studies in each subgroup, we have emphasized in the discussion that results should be interpreted with caution.
Retention Method Discussion:
- Unclear data on cemented vs. screw-retained outcomes; two studies did not report retention type.
R.: We acknowledge that the lack of consistent reporting on retention types in some studies presents a limitation. Despite this, we conducted and included a meta-analysis on prosthetic complications stratified by retention method (screw-retained vs. cemented). Additionally, we revised the meta-analysis on retention loss to reflect this stratification. These updates are now incorporated into the revised version of the manuscript, and caution in interpretation is advised due to the small number of studies.
Concerns raised regarding short (<8 mm) and narrow-diameter implants (<3.5 mm) supporting cantilevers.
- You have to highlight the need for standardized definitions and separate analyses for short/narrow implants; consider recommending caution in discussion.
R.: We have expanded the discussion on the use of narrow-diameter implants in cantilever prostheses, reinforcing the contraindications based on the cases of fracture reported in the literature included in our review. Regarding short implants, as noted in the manuscript, more clinical studies are required to adequately assess their performance in this context.
GRADE Assessment Details:
- Evidence rated low quality, but summary of GRADE judgments is located only in the Supplementary Material. Include an abridged GRADE summary table in the main manuscript for reader clarity.
R.: Thank you for your helpful observation. In response, we have included an abridged GRADE summary table in the main manuscript to enhance clarity and transparency for the reader. This new table has been added as Table 2 and summarizes the certainty of evidence for the main outcomes according to the GRADE approach. The full GRADE assessment remains available in the Supplementary Material for further reference.
Minor Comments:
- Define all abbreviations at first use (e.g., ISPDCs).
R.: All abbreviations were reviewed and are now fully defined at first use. For tables requiring abbreviation for simplification, full definitions are included in the legends.
- Provide exact p-values rather than ‘p > 0.05’ when possible.
R.: Exact p-values have been included throughout the text whenever available, except for values lower than 0.01.
In the Introduction, streamline background citations to avoid redundancy.
R.: The introduction section was revised to eliminate redundancy and improve citation relevance.
This manuscript offers valuable insights into ISPDC performance in the posterior region. Addressing heterogeneity through additional analyses, stratifying by material and retention method, and enhancing clarity will significantly improve the manuscript’s impact and suitability for publication in Materials (MDPI).
Language and Clarity:
- Several typographical inconsistencies (e.g., missing spaces, superfluous line breaks).
- Thorough language editing to improve readability and ensure compliance with MDPI formatting, or get the help of a Native speaker.
R.: We are grateful for your suggestions, which significantly contributed to the refinement of our manuscript. Additionally, a comprehensive English language review was conducted to improve clarity and adherence to academic standards.
Reviewer 3 Report
Comments and Suggestions for Authors
In relation to the paper mentioned above, it is an interesting investigation, although some circumstances and drawbacks must be addressed before considering to be published in this journal. So major changes are considered before going forward.
- Commercial names as Rayyan® must be correctly mentioned, mention the company name, city and country every time it is mentioned in the manuscript.
- Different languages have been included in the search, but no mention about the language distribution of the final included articles. Please descriptive it accordingly.
- Different study design has been included in the search criteria. From the 11 studies included, 10 was used for the meta-analysis, which means all these 10 met the characteristics to be considered for this analysis. Please justify how can perform a high-quality meta-analysis with only 2 RCT. A professional statistician must review this study meta-analysis and report a final evaluation.
- Different restoration materials have been included in the studies included, but no consideration or explanation was found regarding the influence and the predisposition of any primary or secondary objectives in relation to the restorative material.

Author Response
REVIEWER III –
In relation to the paper mentioned above, it is an interesting investigation, although some circumstances and drawbacks must be addressed before considering to be published in this journal. So major changes are considered before going forward.
R.: We thank Reviewer 3 for the critical review and for highlighting important aspects that required improvement. We acknowledge that major revisions were necessary, and we have carefully addressed all comments provided. The corresponding modifications in the manuscript are highlighted in pink for clarity and ease of identification.
- Commercial names as Rayyan® must be correctly mentioned, mention the company name, city and country every time it is mentioned in the manuscript.
R.: We apologize for the error, the modification was made to the methodology.
- Different languages have been included in the search, but no mention about the language distribution of the final included articles. Please descriptive it accordingly.
R.: All studies included in the final sample were in English. We revised the text in the methodology section for clarification (second paragraph).
- Different study design has been included in the search criteria. From the 11 studies included, 10 was used for the meta-analysis, which means all these 10 met the characteristics to be considered for this analysis. Please justify how can perform a high-quality meta-analysis with only 2 RCT. A professional statistician must review this study meta-analysis and report a final evaluation.
R.: We sincerely appreciate your valuable feedback. In response to your observation, we would like to highlight that, according to the recommendations of the Cochrane Consumers and Communication Review Group, it is acceptable to conduct a meta-analysis with only two studies, provided that they can be meaningfully combined and yield sufficiently similar results. As stated by Ryan R. and colleagues, “two studies is a sufficient number to conduct a meta-analysis, as long as those two studies can be meaningfully combined and their results are sufficiently similar” (Ryan R., Cochrane Consumers and Communication Review Group; McKenzie JE, Brennan SE. Summarizing study characteristics and preparing for synthesis).
Nonetheless, we understand your concern regarding the limited number of studies included in our analysis (n=10), and we acknowledge this limitation explicitly in the discussion section of the manuscript. We also emphasize the need for further research, particularly randomized controlled trials, to strengthen the current body of evidence.
Additionally, we performed subgroup analyses based on study design (randomized clinical trials, prospective cohorts, and retrospective studies) as a strategy to address heterogeneity and provide greater clarity in the interpretation of findings.
Finally, we would like to inform you that the authors have training and prior experience in conducting statistical analyses in systematic reviews and meta-analyses, with previous publications in this field. All data and statistical procedures were carefully reviewed and validated prior to submission. We highlight that the research group has published extensively in this area over the past five years. Below is a selection of relevant peer-reviewed articles authored by the group:
- Vieira FL, Carnietto M, Cerqueira Filho JRA, Bordini EA, Oliveira HFFE, Pegoraro TA, Santiago Junior JF. Intraoral Scanning Versus Conventional Methods for Obtaining Full-Arch Implant-Supported Prostheses: A Systematic Review with Meta-Analysis. Applied Sciences (Basel). 2025;15:533–549.
- Costa MSC, Rosa CDDR, Bento VAA, da Silva Costa SM, Santiago JF, Pellizzer EP, de Almeida ALP. Efficacy of acellular xenogeneic dermal matrix graft in the treatment of multiple gingival recessions: systematic review and meta-analysis. Clinical Oral Investigations. 2024;28:xx–xx.
- Sugio CYC, Garcia AAMN, Kitamoto KAA, Santiago Junior JF, Soares S, Porto VC, et al. Mucoadhesive delivery systems for oral candidiasis treatment: A systematic review and meta-analysis. Oral Diseases. 2024;xx:xx–xx.
- Dallavilla GG, Martins DS, Mamani MP, Santiago Junior JF, Rios D, Honório HM. Prevalence of erosive tooth wear in risk group patients: systematic review. Clinical Oral Investigations. 2024;28:588.
- Assem NZ, Pazmino VFC, Rodas MAR, Caliente EA, Dalben GS, Soares S, Santiago Junior JF, de Almeida ALPF. Bone Substitutes Graft for Regeneration of the Anterior Maxillary Alveolar Process: A Systematic Review. J Oral Implantol. 2023;49:102–113.
- Chappuis-Chocano AP, Venante HS, Costa RMB, Pordeus M, Marcillo-Toala OO, Santiago Junior JF, Porto VC. A systematic review and meta-analysis of the clinical performance of implant-supported overdentures retained by CAD-CAM bars. J Appl Oral Sci. 2023;31:xx–xx.
- Faverani LP, Rios BR, Santos AMS, Mendes BC, Santiago Junior JF, Sukotjo C, et al. Predictability of single versus double-barrel vascularized fibula flaps and dental implants in mandibular reconstructions: A systematic review and meta-analysis of prospective studies. J Prosthet Dent. 2023;15:xx.
- Carneiro C, Santiago Junior JF, Peralta LCF, Neppelenbroek KH, Porto VC. What is the Best Tooth-Supported Attachment System for Distal-Removable Partial Dentures? A Systematic Review with Meta-Analysis. Int J Prosthodont. 2023;37:1–11.
- Nova TV, Leão RS, Santiago Junior JF, Pellizzer EP, Vasconcelos BCE, Moraes SLD. Photodynamic therapy in the treatment of denture stomatitis: A systematic review and meta-analysis. J Prosthet Dent. 2022;21:697–xx.
- Miranda G, de Almeida FT, Gasperini G, Silva BSF, Valladares-Neto J, Santiago Junior JF, Silva MAG. Complications in intraoral versus external approach for surgical treatment of Eagle syndrome: A systematic review and meta-analysis. Cranio. 2022;xx:1–13.
- Lemos CAA, Verri FR, Gomes JML, Santiago-Junior JF, Miyashita E, Mendonça G, Pellizzer EP. Survival and prosthetic complications of monolithic ceramic implant-supported single crowns and fixed partial dentures: A systematic review with meta-analysis. J Prosthet Dent. 2022;22:736–743.
- Prasad S, Faverani LP, Santiago Junior JF, Yuan JCC, Sukotjo C. Attachment systems for mandibular implant-supported overdentures: A systematic review and meta-analysis of randomized controlled trials. J Prosthet Dent. 2022;132:354–368.
- Cruz JC, Martins CK, Piassi JEV, Garcia Junior IR, Santiago Junior JF, Faverani LP. Does chlorhexidine reduce the incidence of ventilator-associated pneumonia in ICU patients? A systematic review and meta-analysis. Med Intensiva (Engl Ed). 2022;22:329–xx.
- Pordeus M, Moreira R, Chappuis AP, Venante H, Santiago Junior JF, Porto VC. Computer-aided technology for fabricating removable partial denture frameworks: A systematic review and meta-analysis. J Prosthet Dent. 2021;xx:30396–xx.
- Biguetti CC, Santiago Junior JF, Fiedler MW, Marrelli MT, Brotto M. The toxic effects of chloroquine and hydroxychloroquine on skeletal muscle: a systematic review and meta-analysis. Sci Rep. 2021;11:xx.
- Costa RMB, Venante HS, Pordeus MD, Chappuis-Chocano AP, Neppelenbroek KH, Santiago Junior JF, Porto VC. Does microwave disinfection affect the dimensional stability of denture base acrylic resins? A systematic review. Gerodontology. 2021;xx:xx–xx.
- Lemos CAA, Nunes RG, Santiago-Junior JF, Gomes JML, Limírio JPJ, Rosa CDDR, Verri FR, Pellizzer EP. Are implant-supported removable partial dentures a suitable treatment for partially edentulous patients? A systematic review and meta-analysis. J Prosthet Dent. 2021;xx:xx–xx.
- Poli PP, de Miranda FV, Polo TOB, Santiago Junior JF, Lima Neto TJ, Rios BR, Assunção WG, Ervolino E, Maiorana C, Faverani LP. Titanium Allergy Caused by Dental Implants: A Systematic Literature Review and Case Report. Materials. 2021;14:5239–xx.
- Chappuis-Chocano AP, Venante HS, Costa RMB, Pordeus MD, Santiago Junior JF, Porto VC. Evaluation of the clinical performance of dentures manufactured by computer-aided technology and conventional techniques: A systematic review. J Prosthet Dent. 2021;xx:xx–xx.
- Souza CA, Taborda MB, Momesso GAC, Santiago Junior JF, Santos PH, Rocha E, Assunção WG. Materials sealing preventing biofilm formation in implant/abutment joints: Which is the most effective? A systematic review and meta-analysis. J Oral Implantol. 2020;46:xx–xx.
- Rodas MAR, de Paula BL, Pazmino VFC, Vieira FFSL, Santiago Junior JF, Silveira EMV. Platelet-rich fibrin in coverage of gingival recession: A systematic review and meta-analysis. Eur J Dent. 2020;14:315–326.
- Momesso GAC, Lemos CAA, Santiago Junior JF, Faverani LP, Pellizzer EP. Laser surgery in management of medication-related osteonecrosis of the jaws: A meta-analysis. Oral Maxillofac Surg. 2020;xx:xx.
- Sales-Peres SHC, de Azevedo-Silva LJ, Bonato RCS, Sales-Peres MC, Pinto ACDS, Santiago Junior JF. Coronavirus (SARS-CoV-2) and the risk of obesity for critically illness and ICU admitted: Meta-analysis of the epidemiological evidence. Obes Res Clin Pract. 2020;xx:xx–xx.
- Sukotjo C, Lima-Neto TJ, Santiago Junior JF, Faverani LP, Miloro MM. Is there a role for absorbable metals in surgery? A systematic review and meta-analysis of Mg/Mg alloy based implants. Materials. 2020;13:3914.
- da Costa RMB, Poluha RL, de la Torre Canales G, Santiago Junior JF, Conti PCR, Neppelenbroek KH, Porto VC. The effectiveness of microwave disinfection in treating Candida-associated denture stomatitis: A systematic review and meta-analysis. Clin Oral Investig. 2020;24:3821–3832.
- de Souza Rendohl E, Mizíara LNB, Pimentel AC, Sendyk WR, Santiago Junior JF, Marão HF. The influence of acetylsalicylic acid on bone regeneration: A systematic review and meta-analysis. Br J Oral Maxillofac Surg. 2020;20:30442.
- Different restoration materials have been included in the studies included, but no consideration or explanation was found regarding the influence and the predisposition of any primary or secondary objectives in relation to the restorative material.
R: As complication rates were often presented globally without specifying the restorative material used, the available data did not support a robust material-specific meta-analysis. Nonetheless, we added a new meta-analysis evaluating global complication rates (including retention loss and prosthetic/abutment complications) stratified by occlusal material. These results are discussed with appropriate caution, given the limited number of studies included. This response is marked in yellow, as this issue was also raised by another reviewer.
We sincerely thank Reviewer 3 for their insights, which have contributed meaningfully to improving the scientific and methodological rigor of our manuscript.
Reviewer 4 Report
Comments and Suggestions for Authors
This review is on a serious and frequently debated topic in implant prosthodontics, The survival and complication rate of implant supported cantilevered fixed partial dentures in the posterior region. However, a few methodologic and interpretative aspects need clarification or refinement:
- The authors use terms like "prosthesis failure," "technical complications" and "biological complications" interchangeably. These terms should be clearly defined in the intro or in the methodology section and used uniformly in the paper. For instance, it's not clear whether loosening of the screws was counted as a failure or a complication in Table 1.
- A few studies in the table do not provide information about prosthesis retention mode (cemented or screw-retained), as well as about the material used for the occlusal surface. I would like to invite the authors to revisit these studies or to make it sure that the data was not available or was omitted in the paper.
- A few sentences are far too formal or clumsy to the point that the readability suffered. For instance, sentences like "the physiological stress levels. remain not fully known" could do without the phrase "not fully." I would suggest language refinement.
- The certainty of the evidence was ranked as “very low” for nearly all the results. However, this drawback seems to have been played down in the discussion. The conclusions might do well to have a more circumspect tone; the clinical recommendations need to have a more circumspect tone.
- Figures 4–7 are informative and well-presented but need more precise captions. For instance, the Figure 6 should explain what “>7 mm” implies practically. Also, do include the GRADE summary of the evidence in the main paper and not in the Supplementary Material.
Why was implant connection type (internal/external) not explored as a variable?
Comments on the Quality of English LanguageThe manuscript would benefit from professional English editing. There are awkward phrases and grammatical issues that hinder readability, especially in the results and discussion sections.
Author Response
REVIEWER IV -
This systematic review and meta-analysis addresses an important clinical topic: the survival and complication outcomes of implant-supported fixed partial dentures with cantilever extensions (ISPDCs) in the posterior region. The study is generally well-structured and follows PRISMA guidelines, and the inclusion of meta-analyses strengthens its findings.
R.: We thank Reviewer 4 for the detailed and constructive feedback, which helped us improve the clarity, depth, and overall quality of our manuscript. All major revisions have been carefully addressed, including refinements in terminology, additional subgroup analyses, and clarification of methodological limitations. The corresponding changes are highlighted in blue throughout the manuscript for easy identification.
However, the manuscript requires major revisions in the following areas:
- Clarity and consistency of terminology: The terms used to describe complications (e.g., "prosthesis failure", "technical complications", "biological complications", etc.) must be consistently defined and used throughout the text. Some confusion exists between failure and complication rates.
R.: We reviewed the entire manuscript and clarified the definitions of all terms related to complications (e.g., “prosthesis failure,” “technical complications,” and “biological complications”). These terms are now consistently defined and used throughout the methodology and results sections.
- Details of included studies: The manuscript lacks clarity in reporting retention types (screw vs. cemented) and prosthetic materials for several included studies. Since these factors could influence outcomes, more complete reporting is essential.
R: We acknowledge the lack of reporting on retention type and prosthetic material in several studies. Nevertheless, we included a meta-analysis on prosthetic complications stratified by retention type (screw-retained vs. cemented), as well as an updated analysis for retention loss. Additionally, we performed a meta-analysis on global complications (including retention loss and prosthetic/abutment complications) stratified by occlusal material. These revisions are marked in yellow since they also address comments raised by other reviewers. Due to the limited number of studies, we recommend cautious interpretation.
- Statistical limitations and bias: Although GRADE is mentioned, the overall certainty of evidence is low due to substantial heterogeneity and the predominance of non-randomized studies. This limitation should be emphasized more clearly in the conclusion.
R.: We appreciate your insightful observation. As suggested, we have revised the Abstract, discussion and Conclusion to clearly emphasize the limitations associated with the overall certainty of evidence. Specifically, we acknowledge that the predominance of non-randomized studies and the substantial heterogeneity across included studies contributed to a low certainty of evidence, as assessed using the GRADE approach. These methodological limitations have been explicitly stated to reinforce a cautious interpretation of the findings and the need for further high-quality randomized clinical trials in this area. Furthermore, we added a summary table of the GRADE assessment, which presents the main domains of certainty and clearly highlights the limitations associated with each outcome.
- Subgroup analyses: While subgroup analyses were performed (e.g., cantilever length), other clinically relevant subgroups such as implant diameter, type of occlusal material, and retention method were not explored, despite being frequently discussed in the discussion.
R.: Thank you for this relevant observation. Due to inconsistent reporting across studies, subgroup analyses for implant diameter were not feasible. However, in response to this comment, we included new subgroup analyses based on study design (RCTs, prospective cohorts, retrospective studies), which contributed to explaining heterogeneity. We also conducted subgroup analyses for all outcomes with high heterogeneity, such as those in Section 3.1.6 (biological complications). Additionally, we performed new subgroup meta-analyses based on retention type and occlusal material, as described above. These paragraphs are in yellow because they also included suggestions from reviewer I. Again, due to the small number of included studies, the interpretation should be approached with caution.
- English language and syntax: The manuscript would benefit from professional English editing. There are awkward phrases and grammatical issues that hinder readability, especially in the results and discussion sections.
R.: We carried out a comprehensive English language review, as suggested, to improve the readability and ensure compliance with the journal’s language standards.
- Figures and tables: Ensure that all figures are of high resolution and labeled correctly. The PRISMA flowchart and forest plots should be self-explanatory with complete captions.
R.: We reassessed the image quality, confirming that all figures meet the journal’s requirements (minimum 1000 pixels in width/height or 600 dpi resolution). Captions have also been revised to ensure completeness and clarity.
- Conclusion and clinical implications: The conclusions could be more precise, highlighting not only the low failure rates but also the relatively high rate of complications and the need for case selection based on cantilever length and prosthetic design.
R.: The Conclusion section was refined to more precisely highlight both the low failure rates and the relatively high rate of complications. We also emphasized the importance of case selection based on cantilever length and prosthetic design.
Comments on the Quality of English Language
The manuscript would benefit from professional English editing. There are awkward phrases and grammatical issues that hinder readability, especially in the results and discussion sections.
R.: We carried out a new English assessment as requested.
Reviewer 4’s comments, which provided valuable guidance to strengthen the manuscript’s clinical and scientific contributions.
Round 2
Reviewer 2 Report
Comments and Suggestions for Authors
Dear Authors,
I have carefully read the newest article form and I think it improved according to my advice: now it is ready for publication.
Thank you and good luck!
Reviewer 4 Report
Comments and Suggestions for Authors
Thank you for considering all my suggestions. Great work done